# Microblog Text Emotion Classification Algorithm Based on TCN-BiGRU and Dual Attention

Yao Qin, Yiping Shi *, Xinze Hao and Jin Liu

School of Electronic and Electrical Engineering, Shanghai University of Engineering Science,
Shanghai 201620, China
* Correspondence: syp@sues.edu.cn; Tel.: +86-18918077898

**Abstract:** Microblog is an important platform for mining public opinion, and it is of great value to conduct emotional analysis of microblog texts during the current epidemic. Aiming at the problem that most current emotional classification methods cannot effectively extract deep text features, and that traditional word vectors cannot dynamically obtain the semantics of words according to their context, which leads to classification bias, this research put forward a microblog text emotion classification algorithm based on TCN-BiGRU and dual attention (TCN-BiGRU-DATT). First, the vector representation of the text was obtained using ALBERT. Second, the TCN and BiGRU networks were used to extract the emotional information contained in the text through dual pathway feature extraction, to efficiently obtain the deep semantic features of the text. Then, the dual attention mechanism was introduced to allocate the global weight of the key information in the semantic features, and the emotional features were spliced and fused. Finally, the Softmax classifier was applied for emotion classification. The findings of a comparative experiment on a set of microblog text comments collected throughout the pandemic revealed that the accuracy, recall, and F1 value of the emotion classification method proposed in this paper reached 92.33%, 91.78%, and 91.52%, respectively, which was a significant improvement compared with other models.

**Keywords:** microblog text; sentiment classification; dual attention; fusion features; TCN-BiGRU; ALBERT

## 1. Introduction

Microblogs have become an important medium for people to communicate and express their emotions, as social networks and smart devices have grown in popularity; thus, research on microblog emotions is of great value [1]. Online social media, as represented by Weibo, is the main platform for the dissemination of current public opinion. Due to the network media's information dissemination depth, speed, and large volume, once false information is formulated into online content, it will quickly form online rumors, seriously affecting the stability of society. In particular, with the outbreak of the new coronavirus (COVID-19) pneumonia, a large number of related epidemic comments have been generated on Weibo [2]. Sentiment analysis of netizens under public health emergencies [3] can better understand the emotional trends of the people, provide a reference for the government to regulate the emotions of netizens during the epidemic, and scientifically and efficiently facilitate the prevention and control of publicity and public opinion.

Sentiment analysis, commonly referred to as opinion mining [4], includes the analysis of text features and user opinions, to determine whether their emotional tendencies are positive or negative. The online sentiment classification task [5] has become an important analysis tool for user reviews on social platforms. There are two classifications in traditional sentiment analysis: sentiment dictionary [6,7], and machine learning [8]. The construction of a sentiment dictionary requires a lot of manpower and time, and the operation is time-consuming and complicated. Methods based on machine learning, such as support vector machines [9], Naive Bayes [10], etc., have an obvious disadvantage in that they ignore context information and cannot extract salient features. With the rise of deep learning [11,12],

methods dependent on deep learning are increasingly used in sentiment sorting duties and have better results, among which the most widely used is the convolution neural network (CNN) [13] and recurrent neural network (RNN). Cheng et al. [14] proposed a global RNN-based sentiment classification method to perform sentiment analysis on Weibo comments, and proved the effectiveness of the method. Due to the vanishing gradient problem of RNNs, He et al. [15] employed LSTM to store words with emotional connotations that were particular to the text, emojis, and other features, to improve text classification. However, CNN models cannot learn text sequence features. Therefore, increasingly, studies have proposed temporal convolutional networks (TCN) [16]. Compared with traditional CNN, this model has better performance in large text and structured prediction. Studies found that combining the respective advantages of the CNN and the bidirectional gated recurrent unit (BiGRU) [17] model can improve classification accuracy. Miao et al. [18] constructed a text sentiment analysis technique based on a hybrid model of CNN and BiGRU and verified that the combination of multiple networks could increase the model's precision. To perform a text sentiment analysis, Yang et al. [19] put forward a two-channel convolutional neural network and showed that the parallel hybrid network model outperformed the single network model in terms of performance. The key text information in the sentiment classification task has an important impact on sentiment polarity, so the introduction of an attention mechanism [20] allows better results for feature extraction. Ma et al. [21] combined an attention mechanism with LSTM, to perform target text sentiment classification work, to improve the classification effect. Cao [22] et al. introduced an attention mechanism utilizing a combination of LSTM and TCN networks, to ensure that the model focused on words related to emotion. Cheng et al. [23] added an attention mechanism to multi-channel CNN and BiGRU for experiments and achieved better classification results.

The above deep learning-based methods perform better than traditional sentiment classification methods, but they still cannot effectively extract deep text emotional features, word vectors cannot dynamically obtain the semantics of words according to context, and they lack the ability to extract key information in the text. In order to further improve the accuracy of text sentiment classification, combined with the characteristics of different models, this paper proposes the following improvement methods: Aiming at the problems of poor representation quality of traditional word vectors, and the too many parameters of the BERT pre-training model and the model being too large, ALBERT was used to analyze text and carry out vectorized representation, which can fully consider the information on the left and right sides of the word, to obtain a deeper vector representation containing contextual semantic information. For the problem of the insufficient feature extraction of single deep neural network models and the lack of attention given to key information, the TCN-BiGRU-DATT model was developed. This model uses a two-path computing structure. One path uses the causal convolution and dilated convolution in the temporal convolutional network TCN to extract the emotional features of words, thereby extracting deep text emotions with temporal features. Another path uses the BiGRU network to learn the context information of the comment text and uses the complementary advantages of TCN and BiGRU to more fully extract the features of the text, as well as introducing an attention mechanism into both paths, so that the model can pay more attention to important words in the comments and the model can accurately identify the emotional polarity of the comment text.

Given the above considerations, this paper takes the text of COVID-19 as the research object and proposes a Weibo text emotion classification model based on TCN- BiGRU and dual attention to analyze the public's emotional state during public health events.

## 2. Related Work

### 2.1. ALBERT Pretraining Model

The traditional static word vectors Word2vec [24], Glove [25], etc. cannot be changed according to the context, and the information coverage is relatively simple, so they cannot effectively express the features of the context words in the review text. The development

of pre-trained language models in the field of NLP, such as ELMO [26], GTP [27], etc., has achieved good results in many NLP tasks. Based on the transformers model and self-attention, and to address the issue of text semantic similarity, Devlin et al. [28] suggested BERT (bidirectional encoder representations from transformers). However, the BERT model includes a lot of parameters and requires a lot of training time. To optimize the model, Lan et al. [29] proposed a lightweight ALBERT (a lite BERT) model, which consists of multiple bidirectional transformer encoders with a self-attention mechanism. Its structure is shown in Figure 1, where the characters in the text sequence are represented as $E_1, E_2, \ldots, E_N$. After training by the multi-layer transformer model, the corresponding text feature vector $T_1, T_2, \ldots, T_N$ is output.

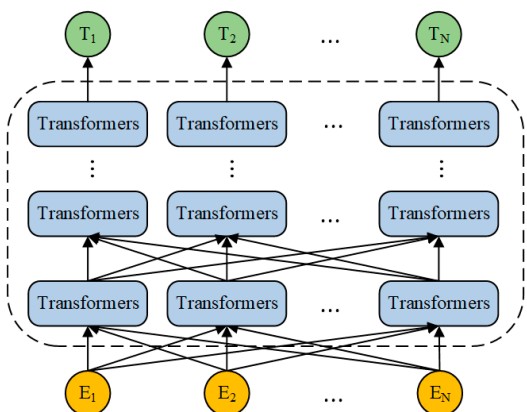

**Figure 1.** ALBERT model.

Among them, the transformer encoder is made up of multiple identical network layer stacks, and every network layer comprises a feedforward layer and a multi-head self-attention mechanism layer. The encoder module for the transformer and the core module is self-attention. The following is the calculation formula:

$$Attention(Q, K, V) = soft\max(\frac{QK^T}{\sqrt{d_k}}) \tag{1}$$

$$MultiHead(Q, K, V) = \\ Concat(head_1, head_2, \ldots, head_h)W^O \tag{2}$$

$$head_i = Attention(QW_i^Q, KW_i^K, VW_i^V) \tag{3}$$

In the formula $Q$, $K$, and $V$ indicate the query matrix, key matrix, and value matrix, respectively. $\sqrt{d_k}$ has the role of regulating and controlling the internal accumulation of $Q$ and $K$. $W_i^Q$, $W_i^K$, and $W_i^V$ represent the weight matrix of the query, the key, and the value vector.

This paper uses the ALBERT pre-training model to generate text word vectors. The ALBERT model is improved on the basis of the BERT model. It uses shared parameters and matrix decomposition technology to reduce parameters, and has no significant impact on the performance of the model. These techniques for reducing parameters can also act as some form of regularization, making the training more stable and the model generalization ability stronger.

### 2.2. TCN Network Model

Temporal convolutional networks perform feature extraction on time scales and are effective in text sentiment analysis [30] and action segmentation [31]. The basic unit of the TCN model is the TCN residual module, which consists of two layers of dilation and causal convolution using a residual connection. For the receptive field, with the increase of the TCN receptive field, the number of layers falls, and the residual network is used to suppress this. It has the advantages of parallel computing, low memory consumption, and

stable gradient. Figure 2a depicts a residual module with two fundamental units, made up of weight normalization, an ReLU activation layer, dropout, and dilated causal convolution of two hidden layers. The network is regularized using weight normalization and dropout, and the residual structure takes the place of straightforward connections between the TCN layers. An illustration of a TCN residual connection is shown in Figure 2b.

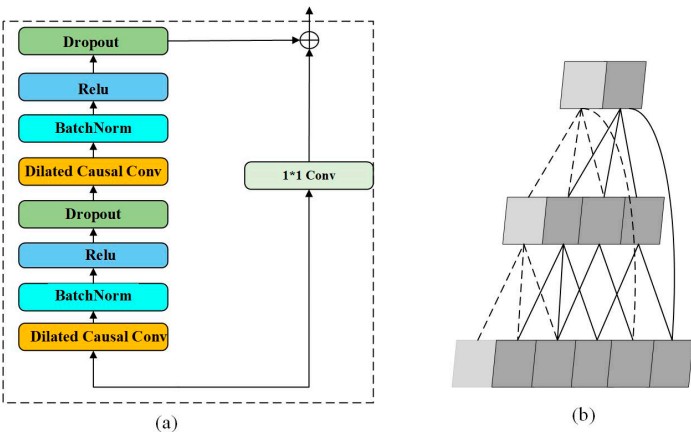

**Figure 2.** TCN residual module. (**a**) residual block containing two basic units. (**b**) example of residual connectivity in TCN.

The TCN model mainly uses convolution calculation to process the text in the time series, conducts a convolution operation on the series by expanding causal convolution, classifies and normalizes the parameters after convolution calculation, and then uses the ReLU activation function to complete the nonlinear calculation. It has a strong time feature extraction ability, and can effectively capture high- and low-dimension hidden features of context sequences. The advantages of TCN are mainly found in the following points: First, the causal convolution introduced by TCN enables TCN to have the ability to process a time series, ensuring that information at historical moments will not be missed, and the introduction of dilated convolution enables the TCN to use fewer network layers. It can also obtain a larger receptive field, ensuring that more timely information can be learned. At the same time, the reduction in the number of layers will reduce the number of parameters, and the memory consumption and amount of calculations will be greatly reduced. In addition, the TCN also introduces a residual module, to address the gradient vanishing issue that may occur during backpropagation, when the number of network layers is large. At the same time, a TCN also has certain shortcomings: causal convolution can only use all the time information before a certain time, and the information of the later time cannot affect the output at the current time, so this ignores the impact of the later sentence on the previous sentence, which is not conducive to the full understanding of the text. In view of this problem, the BiGRU model was used.

## 3. TCN-BiGRU-DATT Model

This paper combines the TCN and BiGRU networks and integrates the attention mechanism, as well as proposes the TCN-BiGRU-DATT sentiment classification model, which takes microblog texts during the epidemic as the input, with the sentiment category as the output. Its structure is shown in Figure 3.

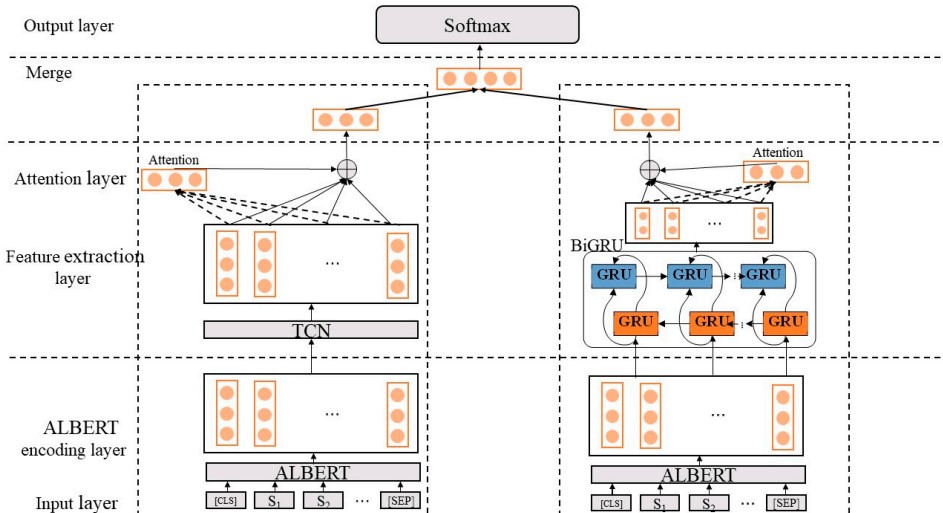

**Figure 3.** TCN-BiGRU-DATT model structure.

The model can be divided into a word vector layer: ALBERT generates a word embedding vector with a contextual semantic feature relationship as a text vector representation; the first channel feature extraction layer includes TCN-ATT feature extraction, which extracts feature information of the remote long-sequence text and local features; the second channel feature extraction layer includes BiGRU-ATT feature extraction, to further understand the text's emotional characteristics, learn the relevant bidirectional semantic components; the output layer can obtain the final sentiment classification results through identifying the output results using Softmax.

### 3.1. Input Layer

In this paper, the ALBERT pretraining model was adjusted to produce the word vector of the text. ALBERT's input is located in the input layer and is a token sequence containing n characters, and the sentence is expressed as $S = s_1, s_2, \ldots, s_n$. If you enter the sentence "Salute the medical staff at the front of the epidemic situation", [CLS] represents the beginning of the sentence and [SEP] marks the end of the sentence.

The vector corresponding to each word input into the ALBERT model is composed of three parts of vectors, namely Token Embeddings, Segment Embeddings, and Position Embeddings, which respectively represent the token value, sentence information, and position information characteristics corresponding to the word. The character feature is $(e_1^t, e_2^t, \ldots, e_n^t)$, the sentence feature is $(e_1^s, e_2^s, \ldots, e_n^s)$, the location feature is $(e_1^p, e_2^p, \ldots, e_n^p)$, and the calculation formula of the input layer is $C_i = (e_i^t + e_i^s + e_i^p)$. Input C into the multi-layer transformer, and output the final word embedded in $X_w = (x_1, x_2, \ldots, x_i) \in R^{L \times d}$, where $L$ represents the length of the sentence and $d$ is the dimension of the word vector.

### 3.2. Feature Extraction Layer

3.2.1. TCN-ATT Feature Extraction Path

Path 1 of this paper contains the TCN network and attention mechanism modules. Based on the ALBERT model, the TCN model is added, to sample and calculate text features based on the feature information output by ALBERT and extract more comprehensive and deep text feature information. Take the vector $h_t$ output from the last hidden layer of the ALBERT model as the input of the TCN model. The precise formula is given below.

$$S_i = Conv(M_i + K_j + b_i) \tag{4}$$

$$\{S_0, S_1, \ldots, S_n\} = LayerNorm(\{S_0, S_1, \ldots, S_n\}) \tag{5}$$

$$\{C_0, C_1, \ldots, C_n\} = ReLU(S_0, S_1, \ldots, S_n) \tag{6}$$

where $S_i$ stands for the status value obtained through time convolution. The word matrix determined via expansion convolution is represented by $M_i$. $K_j$ represents the J-layer convolution core. $b_i$ represents the bias vector. $\{S_0, S_1, \ldots, S_n\}$ means the encoding of the sequence feature vector, $\{C_0, C_1, \ldots, C_n\}$ represents the feature vector obtained after non-linear calculation. After the TCN model is processed, the feature vector $H$ is obtained, and the non-linear transformation is performed to obtain the final output $q$. The following is the accurate formula:

$$H = h_t + \{C_0, C_1, \ldots, C_n\} \tag{7}$$

$$q = HW^{n \times m} \tag{8}$$

Formula (8), $W^{n \times m}$ denotes the parameter matrix of a linear transformation, $n$ represents the dimension of the semantic vector before the conversion, and $m$ represents the dimension after the conversion.

In order to filter out more prominent emotional feature information and improve the classification accuracy, the output matrix $q$ after the TCN convolution operation is introduced into the attention mechanism. The calculation formula is as follows:

$$u_i = \tanh(W_s q_i + b_s) \tag{9}$$

$$\alpha_i = \frac{\exp(u_i)}{\sum\limits_{s=1}^{n} \exp(u_s)} \tag{10}$$

$$F = \sum_{t=1}^{n} \alpha_i q_i \tag{11}$$

In the above formula, $q_i$ represents the feature representation vector learned by the TCN model, $u_i$ is the hidden layer representation of $q_i$ obtained by the attention calculation, $W_s$ represents the weight matrix, $b_s$ represents the bias matrix, and $\alpha_i$ is the normalization obtained by the Softmax value. The normalized weight value represents the value of the influence of the vector on the classification result and represents the feature vector obtained after the weighted operation, and the vector contains important feature information.

### 3.2.2. BiGRU-ATT Feature Extraction Path

In this paper, path 2 contains the BiGRU network and attention mechanism. The BiGRU network is used for bidirectional processing of text sequences, and the transmission state is controlled by the gate structure to realize the memory function. The feature vector extracted from the ALBERT model is input into BiGRU, to obtain the semantic information from left to right and from right to left. Based on the ALBERT model, more long text sequence feature information can be saved. The vector $h_t$ output from the last hidden layer of the ALBERT model is input into the BiGRU network, as shown below:

$$\overrightarrow{l_t} = \overrightarrow{GRU}(\overrightarrow{l_{t-1}}, h_t) \tag{12}$$

$$\overleftarrow{l_t} = \overleftarrow{GRU}(\overleftarrow{l_{t+1}}, h_t) \tag{13}$$

$$l_t = \overrightarrow{l_t} \oplus \overleftarrow{l_t} \tag{14}$$

In the above formula, $\overrightarrow{l_t}$ represents the forward output result of BiGRU at the time $t$, $\overleftarrow{l_t}$ represents the reverse output result of BiGRU at the time $t$, and $l_t$ represents the output result at the time $t$.

To extract more critical information from the review text, the output of the BiGRU model is input into the attention mechanism, the input state at each moment is weighted by

the attention mechanism, and higher greater weight is assigned to the vector information that affects the classification result. Calculated as follows:

$$z_t = \tanh(W_s l_t + b_s) \tag{15}$$

$$\beta_t = \frac{\exp(z_t)}{\sum\limits_{s=1}^{n} \exp(u_s)} \tag{16}$$

$$V = \sum_{t=1}^{n} \alpha_i l_t \tag{17}$$

The above formula, $z_t$ represents the feature vector obtained by Tanh nonlinear transformation, $\beta_t$ represents the weight value of the classification function, $u_s$ is the weight parameter value of the Softmax function, and $W_s$ and $b_s$ represent the weight vector and the bias term, respectively.

### 3.3. Output Layer

The emotion feature vectors obtained by the dual channels are first fused to construct a new feature vector. In order to simplify the model complexity, the vector splicing method is adopted to splice the obtained feature vectors $F$ and $V$, to obtain the final feature vector representing $y$, as shown in Equation (15), where $\oplus$ represents the concatenation of vectors.

$$y = F \oplus V \tag{18}$$

Finally, the fused sentiment feature vector is input into the Softmax classifier, to obtain the final predicted sentiment classification probability value of the model, which is defined as follows:

$$O = Softmax(W \cdot y + b) \tag{19}$$

Formula (19) represents the output sentiment classification probability value, $W$ is the weight matrix, $b$ represents the bias vector, and the value of the category to which the output sentiment belongs determines the polarity of the text's sentiment.

## 4. Experiment and Evaluation

### 4.1. Environment and Analysis for Experiments

The experiment was run on the Windows 10 operating system. The CPU was an AMD Ryzen 75800H with Radeon Graphics 3.20 GHz, the experimental programming language was Python 3.7.3, the development tool was Pycharm2020, and the network model was implemented using the TensorFlow1.15.2 framework.

Since the outbreak of COVID-19, it has become the focus of people's attention, deeply affecting everyone's production and life, and it has not ended yet. The 26th National Retrieval Academic Conference (CCIR2020) carried out an evaluation activity of "Internet users' emotion recognition during the epidemic" in 2020, to help understand the Internet users' feelings about COVID-19. The competition organizers collected data based on 230 subject keywords related to "New Coronary Pneumonia", and captured a total of 1 million Weibo data points from 1 January 2020 to 20 February 2020. Our experimental data set was derived from this evaluation activity. The data set contains 100,000 microblogs manually labeled with 1 (positive), 0 (neutral), and −1 (negative) related to the epidemic situation. There were 25,392 positive samples, 57,619 neutral samples, and 16,902 negative samples. The training set and verification sets were divided 8:2, Table 1 displays the sample data set.

**Table 1.** Sample data set.

| Microblog Text | Emotional Label | Number of Samples |
|---|---|---|
| Worth caring and the whole people are united. | 1 | 25,392 |
| Ventilate more and wash hands frequently. | 0 | 57,619 |
| It's too useless. The epidemic is really annoying. | −1 | 16,902 |

In the training process, an insufficient number of training sets will lead to low model accuracy, and an insufficient number of verification sets will lead to a low verification accuracy. Since the amount of data used in this paper reached a relatively large scale, the data set was directly divided. Since the original data set was arranged in a time series, the repetition of topics and words at a similar period of time may be high, so the data set was divided after the data were disrupted. There was an uneven distribution in the original data set, so stratified sampling was adopted after random shuffling.

*4.2. Experimental Parameter Setting*

The specific parameter settings of this experiment are displayed in Table 2.

**Table 2.** Setting up the model parameters.

| Parameter Name | Parameter Value |
|---|---|
| Learning rate | 0.001 |
| Epoch | 8 |
| Optimizer | Adam |
| Dropout | 0.5 |
| ALBERT hidden_size | 768 |
| BiGRU hidden_size | 128 |
| TCN filter_layer | 4 |
| TCN filter_size | (1, 2, 3, 4) |

*4.3. Evaluation Indicator*

In this experiment, the model's evaluation metrics included the accuracy, recall, and F1 value. The F1 value is calculated using the recall rate and precision, and the specific calculation formula is as follows:

$$Accuracy = \frac{T_P + T_N}{T_P + F_N + F_P + T_N} \tag{20}$$

$$Precision = \frac{T_P}{T_P + F_P} \tag{21}$$

$$Recall = \frac{T_P}{T_P + F_N} \tag{22}$$

$$F_1 = \frac{2 * Precision * Recall}{Precision + Recall} \tag{23}$$

Among them, $T_P$ is the proportion of samples that actually match the predictions of positive instances; $F_N$ is the proportion of samples that actually test positive despite being expected to test negatively; $F_P$ is the number of samples that are actually negative and predicted to be positive examples; the number of samples for which a negative example is really predicted is known as $T_N$.

*4.4. Contrast Experiment*

In order to confirm the model's efficacy in the emotion classification task, different contrast experiments were set with the microblog text data set for the evaluation of "Internet users' emotion recognition during the epidemic" in 2020, which is related to the epidemic.

### 4.4.1. Model Training Learning Curve

Different epoch values were set in the experiment. The research model was fit with the number of iterations. For the microblog epidemic data set, the change trend of the three evaluation indicators of the TCN-BiGRU-DATT network model with the number of iterations is shown in Figure 4. The abscissa is the number of iterations per epoch, the ordinate is the evaluation indicator, and the three curves represent the accuracy, recall, and F1 values, respectively. It can be seen from the learning curves of various indicators of the model in Figure 4 that at the beginning of the training, the parameters were initialized and the curve fluctuated greatly. At the eighth epoch, the accuracy, recall, and F1 values reached large values, and the model had trained better parameters. After the eighth epoch, the accuracy, recall, and F1 values of the model slowly increased, indicating that the model had begun to converge, and that the model was gradually fitting and converging on a stable state. Therefore, 8 was selected as the epoch value in this study.

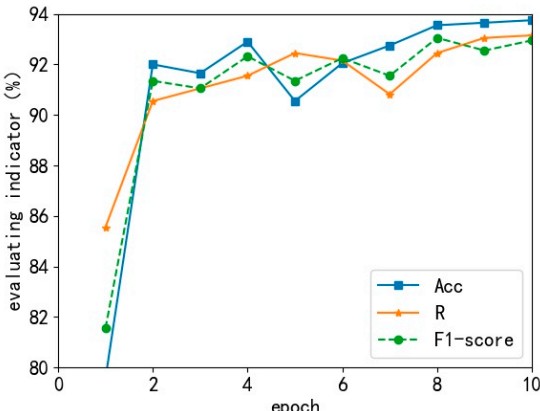

**Figure 4.** Changes in indicators of the TCN-BiGRU-DATT model.

### 4.4.2. Comparison of Different Models

Several sets of comparison models were utilized in this study to validate the classification performance of the model discussed. The models were as follows: (1) TextCNN: text convolutional neural network, which collects local semantic information of a text and has the capability for parallel computing; (2) BiGRU: bidirectional gated recursive unit, this model extracts forward and backward features separately using a GRU, and then combines bidirectional features to obtain context information; (3) TCN: temporal convolutional network, with ReLU as an activation function; (4) FFA-BiAGRU [32]: combining the classification model of attention and gating unit fusion, the attention mechanism and the update gate of GRU are fused to form a hybrid model, to extract important feature information in the text; (5) BiGRU-CNN: a convolutional neural network and bidirectional gated recursive units, are serially connected to extract text features for aspect-level sentiment classification; (6) ALBERT-BiGRU-ATT: the single-path model of this paper; (7) ALBERT-TCN-ATT: the single-path model of this paper.

According to the results in Table 3, since BiGRU can capture feature information in both the front and rear directions, compared to CNN, the acc, recall, and F1 value were better. Compared with the BiGRU model, the three evaluation indicators of the FFA-BiAGRU model were improved, indicating that the addition of the attention mechanism enhanced the model's categorization precision. Comparing the TCN network model with the basic network models of CNN and BiGRU on the epidemic microblog text data, the F1 value, recall rate, and accuracy rate all greatly increased, because TCN has strong timing

properties and can capture long-distance long-range texts, as well as local and global features, to obtain better classification results.

**Table 3.** Comparison results of the different models.

| Model | Acc/% | R/% | F1/% |
|---|---|---|---|
| TextCNN | 84.36 | 83.93 | 84.14 |
| BiGRU | 86.03 | 85.82 | 85.92 |
| TCN | 87.81 | 86.79 | 87.30 |
| FFA-BiGRU | 88.64 | 88.31 | 88.47 |
| BiGRU-CNN | 89.52 | 89.20 | 89.36 |
| ALBERT-BiGRU-ATT | 90.78 | 90.57 | 90.67 |
| ALBERT-TCN-ATT | 91.34 | 91.03 | 90.83 |
| Our Model | 92.33 | 91.78 | 91.52 |

Compared with the BiGRU-CNN model, this paper's model improved the accuracy by 2.81%, the recall rate by 1.52%, and the F1 value by 2.16% on the epidemic microblog data set. Therefore, the parallel network method used in this paper to extract text emotional features was better. Text sentiment features were extracted in the serial manner adopted by the BiGRU-CNN model.

It is clear from Table 3 that, compared with the single-channel ALBERT-BiGRU-ATT and ALBERT-TCN-ATT models, the dual-channel model was superior to the single-channel model, and the fusion of the two channel networks played a complementary positive role, enriching the feature vector. Among them, the TCN model performs causal convolution and dilated convolution to extract regional and global textual characteristics to obtain deeper features, and BiGRU extracts bidirectional semantic information features to strengthen the learning sequence information and increase the model's precision.

The model comparison shows that the TCN-BiGRU-DATT network model used in this paper fully utilized TCN to obtain deep text sentiment feature extraction through causal convolution and dilated convolution, and BiGRU could extract bidirectional information features; and through lightweight ALBERT, the pre-trained language model improved the semantic expression ability of the text, demonstrating the excellence of the approach used in this paper.

4.4.3. Comparison of Different Word Vector Extraction Effects

This experiment converted text into word vectors through the Word2vec, ELMO, BERT, and ALBERT models, and then use TCN-Attention and BiGRU-Attention dual path neural networks for training. The laboratory test results are displayed in Table 4.

**Table 4.** Contrasting various word vector models.

| Model | Acc/% | R/% | F1/% |
|---|---|---|---|
| Word2vec | 86.79 | 85.86 | 86.32 |
| ELMO | 89.74 | 88.96 | 89.35 |
| BERT | 91.45 | 90.21 | 90.76 |
| ALBERT | 92.33 | 91.78 | 91.52 |

From Table 4, the conclusion is that the ELMO, BERT, and ALBERT pretraining models had a better extraction performance in terms of accuracy, recall, and F1 value compared to the conventional static word vector Word2vec. Since the Word2vec word vector language model is static, it cannot obtain the vector representation of the text according to the context, and cannot effectively express the characteristics of the comment text. The word vectors obtained by the pre-training model are all dynamic, and there are differences between different word vectors and contexts. Dynamic association is used, so the extraction effect is better. Since ELMO uses the LSTM language model to obtain word vectors, while BERT

and ALBERT use a more powerful transformer encoder than LSTM to obtain word vectors, BERT and ALBERT have a better performance in extracting word vector features. The BERT model has powerful text representation capabilities. The ALBERT model optimizes the BERT model, while reducing the number of parameters and increasing the model's training efficiency; it had the most accurate recall and F1 values on the epidemic microblog text data set.

### 4.4.4. The Influence of the Attention Mechanism on the Classification Results

Four groups of comparative experiments were run based on the model in this research: (1) TCN+BiGRU removed both the attention in the original network and retained the backbone network model for training; (2) TCN-Attention+BiGRU removed the attention mechanism in the BiGRU model and retained TCN-Attention and BiGRU models for training. (3) TCN+BiGRU-Attention removed the attention mechanism from the TCN model and retained the TCN and BiGRU-attention network models for training. (4) TCN-BiGRU-DATT (the model in this paper). The variation trend of the F1 value in the obtained experimental results is shown in Figure 5.

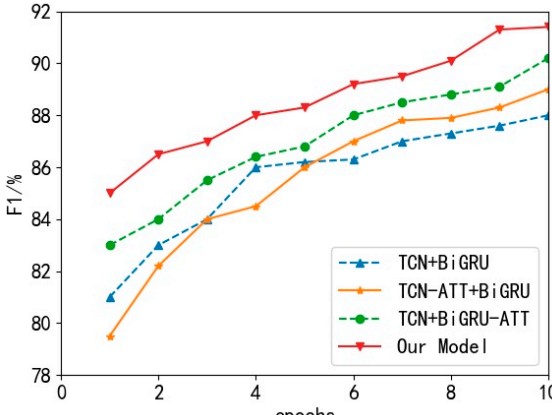

**Figure 5.** F1 value change of the different attention mechanisms.

From the results shown in Figure 5, it was found that, on the epidemic microblog text data set, the emotional efficiency of the three models utilizing the attention mechanism was greatly improved compared to not introducing the attention mechanism. This shows that the introduction of an attention mechanism can capture key feature information and enhance the sentiment classification model's performance. Compared with the TCN-Attention+BiGRU and TCN+BiGRU-attention models using the one-way attention mechanism, the model in this paper had the largest F1 value on the epidemic microblog text data set, indicating that the combination of the two attention mechanisms was more effective. The association of semantic information was completed and made up for the loss of information. Therefore, the TCN-BiGRU-DATT model utilizing the dual attention mechanism in this study was superior, as shown by the fact that the performance of the model using the single attention mechanism was worse than that of the model using the dual attention mechanism.

### 4.5. Analysis of Experimental Results of the Microblog Text Data

The improved ALBERT-TCN-BiGRU classification algorithm was applied in this paper to an unlabeled Weibo comment data set after training, randomly select 100,000 comments from the unlabeled 900,000 comments, to test the classification of unclassified samples, and to check the accuracy of the classification results in combination with the date of publication of the specific method.

It can be seen from Figure 6 that the overall number of Weibo comments began to rise sharply around 18 January 2020. At this time, this was related to topical issues in

society: On 18 January, Professor Zhong Nanshan clarified that the COVID-19 existed in the population. The possibility of transmission was reported, clearly pointing out that wearing a mask is helpful to block the spread of the virus, which led to masks being snapped up, and a large number of masks became out of stock. Wuhan officially "closed the city" on 23 January; on 25 January, Wuhan Huoshenshan Hospital began construction and Leishenshan Hospital finalized its design plan. The occurrence of these important social events coincided with the trends in the number of Weibo comments predicted by the model during this period.

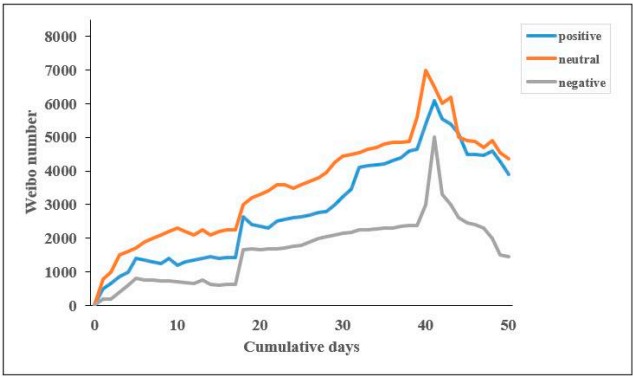

**Figure 6.** Distribution of reviews over time.

On 9 February 2020, the total number of topics on Weibo reached its peak. At this time, social hotspots included Dr. Li Wenliang, who had been the first to release information about the new coronavirus, passed away on the evening of 7 February. There were mostly texts in memory of Li Wenliang on Weibo. In addition, it can be seen from the figure above that there was a slight drop in positive comments on Weibo on February 7. With this being consistent with the death of Dr. Li Wenliang and the increase of pessimistic voices in society. At the same time, the data performance of the ALBERT-TCN-BiGRU classification algorithm proposed in this paper on the unlabeled sample data set can be roughly judged according to the two important time nodes. The number of neutral comments accounts for about 50%, while the number of positive comments increased significantly in February. It can be seen that when local governments began strict prevention and control measures, the people's confidence in the campaign was consolidated, and the people's enthusiasm for fighting the epidemic increased significantly.

## 5. Conclusions

This paper aimed to classify the sentiment of Weibo text comments during the epidemic, so as to obtain the emotional tendency of netizens. A sentiment classification method for microblog texts based on TCN-BiGRU-DATT was proposed. Through the lightweight ALBERT pre-training model, the semantic expression ability of the text was improved. Combining the two-way network model of TCN and BiGRU allowed combining the characteristics of the TCN to obtain deep text emotional features through causal convolution, expansion convolution, and BiGRU's two-way information feature extraction, as well as to integrate the features extracted by the two networks into the attention mechanism and to give the key words learnt more weight. The experimental results showed that the classification performance of the model on the microblog text data set was better than other models, which proved the superiority of the method in this paper.

Due to the limitations of computing resources, personal, and other factors, although this paper made some conclusions, there are still many deficiencies. In the process of data preprocessing, the processing of missing data and sample imbalance is more suitable for experimental data sets and does not conform to the principles of actual application scenarios or a targeted analysis of text quality. In the future, the accuracy of the final evaluation model performance can be improved by modifying the loss function, such as by

using focal loss or by adjusting the weights of samples in different categories. The final results of the experiment in this paper were good but not good enough. It may be that the bias of some specific sentences is not obvious, which leads to the training effect being poor. In the future, the text will be carefully analyzed to evaluate the text quality.

In follow-up work, more channels will be introduced based on the pretraining model, and more detailed research will be performed on emotion analysis.

**Author Contributions:** Conceptualization, Y.Q. and Y.S.; formal analysis, Y.Q. and J.L.; investigation, Y.Q and J.L; data curation, X.H.; Methodology, Y.S.; writing—review and editing, Y.Q., Y.S., J.L.; project administration, Y.S. All authors have read and agreed to the published version of the manuscript.

**Funding:** National Natural Science Foundation (61701296).

**Institutional Review Board Statement:** Not applicable.

**Informed Consent Statement:** Not applicable.

**Data Availability Statement:** Not applicable.

**Conflicts of Interest:** The authors declare no conflict of interest.

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
