# Peer review of "Microblog Text Emotion Classification Algorithm Based on TCN-BiGRU and Dual Attention"

_information, doi:10.3390/info14020090_

Round 1

Reviewer 1 Report

Research Summary

This article reports a proposition of a model based on temporal convolutional networks (TCN), bidirectional gated recurrent unit (BiGRU), and dual attention (DATT) to apply emotion classification to microblog posts. The model uses a light form of bidirectional encoder representations from transformers (ALBERT) to generate the words vectors with contextual semantic features relationships to be processed by two feature extraction channels: one using TCN and another BiGRU ensuring a better understanding of the emotional characteristics in the texts. Each channel works with an attention layer and in the output layer, the Softmax classifier was used to obtain the posts' texts related sentiments.

Major Strengths

1.       In my opinion, the authors did a good job describing their proposed methods. The level of detailing ensures understanding of each step in the involved process, which guarantees reproducibility.

2.       More than half of the references are from 2019 to 2022, ensuring that the authors sought current knowledge to support their proposal without leaving aside the theoretical contributions prior to this period.

3.       According to the tests applied, the model performed better than other pre-existing models for similar purposes.

4.       Another characteristic that I think is quite relevant in texts on propositional studies: the authors were objective and direct in the writing of their text.

Reviewer's Suggestions

1.       A suggestion that I make initially, as I consider it to be quite significant: please develop a section on the theoretical and practical implications based on the proposal made. (a) For the theoretical implications, it is interesting to point out how the article advances knowledge about classifying emotions, sentiments, and opinions. (b) For the practical implications, I think it is interesting to highlight application areas that can benefit well from this type of proposal. Some recent references to assist in the construction of practical implications:

·         Kothamasu, LA, Kannan, E. Sentiment analysis on twitter data based on spider monkey optimization and deep learning for future prediction of the brands. Concurrency Computat Pract Exper. 2022; 34(21): e7104. doi:10.1002/cpe.7104 (with an application in Marketing/Branding).

·         Baidyanath Biswas, Arunabha Mukhopadhyay, Sudip Bhattacharjee, Ajay Kumar, Dursun Delen. A text-mining based cyber-risk assessment and mitigation framework for critical analysis of online hacker forums. Decision Support Systems, Volume 152, 2022. https://doi.org/10.1016/j.dss.2021.113651. (the authors proposed a text-mining framework for cyber-risk assessment, which also uses sentiment analysis).

·         de Carvalho, Victor Diogho Heuer, Thyago Celso Cavalcante Nepomuceno, Thiago Poleto, Jean Gomes Turet, and Ana Paula Cabral Seixas Costa. 2022. "Mining Public Opinions on COVID-19 Vaccination: A Temporal Analysis to Support Combating Misinformation" Tropical Medicine and Infectious Disease 7, no. 10: 256. https://doi.org/10.3390/tropicalmed7100256 (with one application of opinion analysis to support misinformation combat in the infodemic context related to the COVID-19 pandemic).

·         Chen CWS, Fan T-H. 2022.Public opinion concerning governments' response to the COVID-19 pandemic. PLoS ONE 17(3): e0260062. https://doi.org/10.1371/journal.pone.0260062 (this one is also on the COVID-19 theme, but regarding the opinion analysis about political issues).

2.       If it is possible presenting the proposed model's confusion matrix and learning curves (training and testing), I would recommend it to the authors. Especially the learning curves would support understanding the effect of possible overfitting in the authors' proposal.

3.       The receiver operating characteristic curve (ROC) and the area under the curves (AUC) are also welcome to support the understanding of the correctness of the proposal in the classification.

4.       The samples that make up the corpus used are unbalanced. Did this have any effect on training and, consequently, on the classification process? It is important to make a comment containing the authors' assessment of this, perhaps after Table 1.

5.       What are the authors' limitations, challenges, and technical difficulties in implementing their proposal? It is also necessary to present this, being transparent to ensure the reproducibility of the study. I believe that a short section, with two or at most three paragraphs, inside the conclusion can fulfill the objective of presenting this.

6.       The last part of the current conclusions presents a direction for further work; however, the authors can develop this further as the last subsection of "Future Work," closing the conclusions.

Final Considerations

The text is well written, and, in my opinion, the work is well developed, lacking only a few elements to be definitively accepted, as I pointed out above.

Reviewer 2 Report

The paper proposed an approach for sentiment classification. The contribution itself seems to be adding an attention mechanism, which a rather incremental. It is not clear why TCN-BiGRU-DATT model is focus only on the pandemic, it can be adapted to be a general purpose sentiment analysis method?

The introduction section needs to re-written focusing in the motivation and contribution of the work. It enumerates a number of deep-learning related approaches, but never clarified which are their limitation and advantages of the new proposal. Likewise, the Related Works section suffers from the same problem. It is more a background section on two ALBERT and TCN than an actual discussion of related literature. The paper have to discuss the state-of-the-art in the subject, clarify the limitations of previous works and then highlight the novelty of the contribution. It also needs to clarify the delta respecting TCN-BiGRU model.

Experimental results needs to be reported with more detail. First, data is not properly described. There are many aspects that are not mentioned. For example, how was the data labeled? How many annotators were involved? level of agreement, etc. It would be important to state whether the dataset will be available for reproducibility and research purposes in the future. Table 1 columns are confusing, there are many examples of the same microblog text? Also, experiments and the comparison with other approach lacks of detail. Is the data used for experiments (in sect 4.4) the same described in section 4.1? A single holdout partition 80:20 was made? (if so, why not cross-validation), are differences in the metrics statistically significant?. This section should be carefully revise to include all the details to fully validate the approach as well as enable reproducibility.

Author Response

Point 1: About the Introduction and Related Work Section

Response 1:

Thank you for your valuable comments. For the introduction part, in addition to the background and literature introduction of related sentiment analysis, the motivation and contribution of the work have been added. The improvements are as follows:

Most current sentiment classification methods still cannot effectively extract deep text emotional features, traditional word vectors cannot dynamically obtain the semantics of words according to the context, and lack the ability to extract key information in the text, resulting in classification deviations. In order to further improve the accuracy of text sentiment classification, combined with the characteristics of different models, this paper proposes corresponding improvement methods. The specific research contents are as follows:

(1) In view of the poor quality of the traditional word vector representation and the excessive parameters of the BERT pre-training model, the model is too large, the lightweight ALBERT model is used to train the word vector as the word embedding layer, and ALBERT is used to vectorize the text. It can fully consider the information on the left and right sides of the word, so as to obtain a deeper vector representation containing contextual semantic information.

(2) Aiming at the problem of insufficient feature extraction and lack of attention to key information of a single deep neural network model, the TCN-BiGRU-DATT model was designed and implemented. The causal convolution and expansion convolution of the words are used to extract the emotional features of words, so as to extract the deep text emotional information with temporal features; another channel uses the BiGRU network to learn the context information of the comment text, and utilizes the complementary advantages of TCN and BiGRU to more fully extract The characteristics of the text, and the attention mechanism is introduced in both channels, so that the model can pay more attention to the important words in the comments, so that the model can accurately identify the emotional polarity of the comment text. (line73-93).

In addition to introducing the basic background of TCN and ALBERT, the relevant work part adds the advantages of choosing them, and the improvements are as follows:

The advantages and limitations of TCN are explained in part: Through the above description, it can be seen that the advantages of TCN are mainly concentrated in the following points: First, the causal convolution introduced by TCN enables TCN to have the ability to process time series, ensuring that the historical moment information is not consistent. will be missed. The introduction of dilated convolution enables TCN to obtain a larger receptive field with fewer network layers, ensuring that more time information can be learned. At the same time, the reduction in the number of layers reduces the number of parameters and memory consumption. And the amount of calculation will be greatly reduced; in addition, TCN also introduces a residual module to solve the problem of gradient disappearance that may occur during backpropagation when the network layer is deep. But at the same time, TCN also has certain shortcomings: causal convolution can only use all the time information before a certain time, and the information at the later time cannot affect the output at the current time, so it ignores the impact of the later sentence on the previous sentence, which is not conducive to the text full understanding. In view of this problem, the BiGRU model is used to solve it. (line148-161).

The ALBERT part has been improved: For the BERT model with too many parameters and the model is too large, this paper uses the ALBERT pre-training model to generate text word vectors. The ALBERT model is improved on the basis of the BERT model, using shared parameters and matrix decomposition technology to reduce parameters, replace NSP (Next Sentence Prediction) with SOP (Sentence Order Prediction), factorization and cross-layer parameter sharing can effectively reduce the parameters of the model, and have no significant impact on the performance of the model. These techniques for reducing parameters are also It can act as some form of regularization, making the training more stable and the model generalization ability stronger. (line125-133).

Point 2: Dataset related issues

Response 2: Thank you for your valuable comments, and I will explain the data set problem you raised: The data set in this article comes from the evaluation activity of "Recognition of Internet Users' Emotions During the Epidemic" carried out by the 26th National Conference on Retrieval (CCIR2020) in 2020. The competition is a scientific and technological battle-big data public welfare challenge held on the Qingbo public opinion analysis system platform in March 2020 by the Beijing Municipal Bureau of Economy and Information Technology, the CCF Big Data Expert Committee, and the Information Retrieval Professional Committee of the Chinese Information Society of China. One of the competition questions. The competition organizers collected data based on 230 subject keywords related to "New Coronary Pneumonia", and captured a total of 1 million Weibo data from January 1, 2020 to February 20, 2020. Among them, 100,000 items have been manually labeled, and the other 900,000 items are unlabeled data sets. This data set is available for download on the public Internet. The experiment in this paper is mainly based on the labeled data. Use various models for sentiment classification training, and observe the final classification effect on the test set. Table 1 shows some samples in the 100,000 labeled datasets. After Table 1, the imbalance problem of the data set is explained as follows: Since the original data set is arranged in time series, the repetition of topics and words in a similar period of time may be high, so the data is divided into data sets after being disrupted. There is a problem of uneven distribution in the original data set, so stratified random sampling is adopted after random shuffling. (line288-294).

Point 3: About the experimental part

Response 3: Thank you for your valuable comments. The data set used in Section 4.4 of the experiment is the same data set used in Section 4.1, which is explained at the beginning of Section 4.4. This paper uses the two columns of marked Weibo Chinese comment content and emotional tendency. In the training process, insufficient number of training sets will lead to low model accuracy, and insufficient number of verification sets will lead to low verification accuracy. Since the amount of data used in this paper has reached a relatively large scale, the data set is directly divided, and the data set is divided according to the ratio of 8:2 between the training set and the test set. The three evaluation metrics in this paper can well show the classification results. In the experimental part, different models were selected and multiple groups of different comparative experiments were carried out, which fully covered the proposed points and were presented in the experiments. The data sets used are all public microblog epidemic data sets. The advantages of the model in this paper have been verified through experiments. At the same time, relevant supplements have been made to the experimental part, and two experiments have been added. One section shows the model training under different iterations The learning curve better shows the training effect and fitting status of the model(line317-331); the other section is about the theoretical and practical impact, which can more clearly present the topics in the specific time period and the number of Weibo comments predicted by the model. The trend during this period is consistent. (line417-448).

Round 2

Reviewer 2 Report

The revised version of the  paper addressed most of my previous concerns. Although the paper empathizes on COVID-19 as the research object and proposes a Weibo, it would be good to claify why it can not be used in other contexts or its limitations in such case. There are some typos and English mistakes  For example, ¨These The techniques¨ in page 3, among others.

Author Response

Point 1: About model applicability

Response 1:

    Thank you for your valuable comments. The model proposed in this paper is a general text emotion classification model, which is applicable to any other text emotion classification method. The starting point of this study is to count a large number of comments on microblogs during the epidemic, analyze their emotional tendencies, and then mine the emotional tendencies of most user comments. The selected application scenario is the microblog comment text during the epidemic. In preparation for the big paper, I also used this model for experiments. I used different data sets for comparative experiments. In addition to the COVID-19 review text data set, I also used the Chinese corpus of hotel reviews collected by Dr. Tan Songbo of the Institute of Chinese Academy of Sciences, which is also an open data set. The experimental results show that the proposed model has high comprehensive performance, good classification results have also been achieved, and the models proposed in this paper are applicable to different web comment text data sets.

Point 2: About English spelling

Response 2: Thank you for your valuable comments. In response to the English spelling mistakes you raised, I have carefully checked the full text and made corresponding revisions.
